# Presence of Free-living *Acanthamoeba* in Loa and Salado Rivers, Atacama Desert, Northern Chile

**DOI:** 10.3390/microorganisms10122315

**Published:** 2022-11-23

**Authors:** Camila Salazar-Ardiles, Alexander Pérez-Arancibia, Leyla Asserella-Rebollo, Benito Gómez-Silva

**Affiliations:** 1Departamento Tecnología Médica, Health Sciences Faculty, Universidad de Antofagasta, Antofagasta 1270300, Chile; 2Laboratory of Biochemistry, Biomedical Department, Health Sciences Faculty, Centre for Biotechnology and Bioengineering (CeBiB), Universidad de Antofagasta, Antofagasta 1270300, Chile; 3Escuela de Tecnología Médica, Facultad de Salud, Universidad Santo Tomás, Santo Tomas 083067, Chile

**Keywords:** free-living amoebae, *Acanthamoeba*, Atacama Desert, Loa River, Salado River, Chiuchiu Pond, Tebenquiche, Salar de Atacama

## Abstract

Substantial knowledge has accumulated on the microbiome of the hyperarid Atacama Desert during the last two decades; however, information on Atacama free-living amoebae (FLA) is limited and increasing efforts are required. FLA are polyphyletic heterotrophic naked or testate protists that feed on organic matter, fungi, protozoa, and bacteria and may disseminate infections. Amoebae in Chile are represented by 416 taxa and 64 genera, and 29 taxa have been identified in arid shrub lands at the southern limit of the Atacama Desert, and *Acanthamoeba* are present in all the country’s regions. To expand our knowledge and to contribute to the biogeographic distribution of Atacama FLA, we report the dominant presence of members of the genus *Acanthamoeba* in water and sediment sampled at the Loa and Salado rivers in the pre-Andean zone of the Antofagasta Region, northern Chile, at sites 2500 m above sea level. We expect these observations and preliminary evidence of FLA presence in other wetlands (Chiuchiu, Tebenquiche) in this region to be incentive for further exploration of Atacama amoebae.

## 1. Introduction

Free-living amoeba (FLA) are unicellular eukaryotes and a polyphyletic group of cosmopolitan heterotrophic protists involved in recycling of organic matter, detritus, and nutrients [1,2,3]. Based on molecular phylogeny, amoebae are included in the cluster Amoebozoa Lüe, 1913, emend. Cavalier-Smith, 1998, with 255 known genera [4]. FLA can be found as naked or testate amoebae, in air, water, and soil environments, and some FLA are natural reservoirs of pathogenic microbial endosymbionts enclosed within the cysts or the trophozoites [1,5,6]. In Chile, testate amoebae are represented by 416 taxa and 64 genera; they are shelled protists better adapted than naked amoeba to tolerate changes in their environment [7].

Members of the genus *Acanthamoeba* have been identified and associated with public health risks in rural and urban locations and at arid shrub lands at the southern limit of the Atacama Desert [5,7]. There is an increasing scientific attention on *Acanthamoeba* spp. since they are ubiquitous naked free-living microbial predators and opportunistic disseminators of infections whose life cycle includes vegetative growth as trophozoites under favorable conditions and a dormant cyst stage that allows them to survive starvation by forming double-layer cysts [1,8].

The Atacama Desert in northern Chile is a coastal nonpolar hyperarid desert located at the southwest border of South America and it is considered the oldest and driest desert on Earth [9,10,11]. Moreover, Atacama is described as a polyextreme environment for life where the abundance and diversity of microorganisms is limited by major restrictive natural stressors, particularly desiccation and high solar visible and ultraviolet irradiation [10]. Substantial advances have been achieved on the Atacama microbiome [10,11,12,13,14], but scarce information is available on the presence, identification, distribution, diversity, and environmental niches of Atacama FLA [5,7,15], and increasing efforts must be made to expand our knowledge of Atacama amoebae. In this report, we demonstrate the presence and identification of members of the genus *Acanthamoeba* in water and sediment samples from two major rivers in northern Chile at sites located at nearly 2500 m above sea level (asl) at the pre-Andean corridor of the Region of Antofagasta.

## 2. Materials and Methods

### 2.1. Sampling Sites

A total of seven environmental samples were retrieved during June 2019 from Loa and Salado rivers at the pre-Andean area, Region of Antofagasta, in northern Chile (Figure 1 and Table 1). Duplicates of 500 mL of water (10–20 cm depth) and 250 g of sediment samples (upper 5 cm) were obtained from Loa River and Salado River (Figure 1 and Table 1).

### 2.2. FLA Growth and Isolation

Water samples were centrifuged at 2500 rpm for 5 min at room temperature (rotor SS-34, Sorvall RC5B Plus), and the pellets were resuspended in 100 uL of saline Page’s solution and plated on non-nutritive agar medium (NNA; Ryvex LLC, Miramar, FL, USA) supplemented with Page’s solution, seeded with live *E*. *coli* (ATTC 25922) previously grown in Luria Bertani medium. Sediment samples (5 g) were homogenized and divided into four parts, and one of them was suspended in 10 mL of Page’s solution and handled as were the water samples. Plates were incubated at room temperature for 3–5 days until trophozoite growth was confirmed by daily microscopic observation [16]. Agar pieces (2 cm^2^) containing trophozoites were transferred to fresh plates to isolate and purify growing amoebas. 

### 2.3. DNA Extraction

Plates with high trophozoite content received 4.0 mL of Page’s solution, scraped to release adhered trophozoites, and after shaking at 100 rpm for 15 min, the suspensions were recovered in 15-mL conical tubes, and the pellet was recovered after centrifugation at 2500 rpm (SS-34, Sorvall RC5BPlus) for 10 min, at room temperature. Pellets were resuspended in 50 µL of Page´s solution and total DNA was extracted using GeneJETTM genomic DNA purification kit (ThermoFisher Sci., Santiago, Chile), following the manufacturer’s instructions, and stored at −20 °C.

### 2.4. 18S rDNA Gene PCR

The PCR reaction mix (50 uL) was incubated for 7 min at 95 °C, followed by 35 cycles of 1 min at 95 °C, 1 min at 60 °C, and 2 min at 72 °C [16]. PCR was conducted with specific primers for amoebae: Universal Primers Gen 18S rDNA (F: CGGTAATTCCAGCTCCAATAGC, R: CAGGTTAAGGTCTCGTTCGTTAAC; with an expected amplicon size between 700–900 bp) [16,17,18]; JDP (F: GGCCCAGATCGTTTACCGTGAA, R: TCTCACAAGCTGCTAGGGAGTCA) [16]; BAL (F: CGCATGTATGAAGAAGACCA, R: TTACCTATATAATTGTCGATACCA; with an amplicon length between 1000–1200 bp) [19]; ITS (F: AACCTGCGTAGGGATCATTT, R: TTTCTTTTCCTCCCCTTA; with an expected amplicon of 300–450 bp) [20]; and NFITSFW (F: TGAAAACCTTTTTTCCATTTAC, R: AATAAAAGATTGACCATTTGAAA; with an amplicon length of 300–450) [21].

### 2.5. Amplicon Sequencing and Analyses

After agarose electrophoresis, amplicons were recovered with UltraClean^®^ 15 DNA Purification Kit (ThermoFisher Sci., Santiago, Chile), following the manufacturer’s instructions. Amplicons were sequenced at AUSTRAL-Omics Laboratory (Institute of Biochemistry and Microbiology, Universidad Austral, Valdivia, Chile). Phylogenetic analyses were conducted by the maximum-likelihood method based on 18S rRNA gene sequences, using BLASTn program for comparison to NCBI data base. Alignment and phylogeny studies were performed with programs ClustalW and Mega7 [22].

### 2.6. Data Deposition

The sequences are available at NCBI GenBank with the accession numbers OP297695 for Salado sample SRW-1 and OP297696, OP297697, and OP297698 for Loa samples LRSW-3, LRS-3, and LRS-1, respectively. 

## 3. Results

Sampling sites were selected at the Loa and Salado rivers to search for evidence of the presence of FLA in pre-Andean wetlands in the hyperarid Atacama Desert at nearly 2500 m asl in the Region of Antofagasta, northern Chile (Figure 1).

Excepting two Loa samples, all river samples rendered successful FLA growth and showed dominance of the *Acanthamoeba* species (Table 1). Light microscopy images of isolated Loa *Acanthamoeba* spp. are shown as trophozoites or cysts in Figure 2A,B, respectively.

Genus identification for isolated Atacama FLA was based on PCR assays using specific amoebae primers for 18S rDNA gene amplification. To confirm the genus identification of Atacama FLA by PCR, amplicons were sequenced, and the BLAST analyses are summarized in Table 2. The Atacama sequences rendered an 89% to 100% identity with *Acanthamoeba* species registered in the NCBI database.

A phylogenetic tree was constructed with the amplicon sequences from Atacama *Acanthamoeba* spp. (Figure 3). This analysis confirmed that isolates LRS-1 and LRS-3 and isolate SRW-1 SRW-9 from Loa and Salado samples, respectively, would be unambiguously assigned to the genus *Acanthamoeba*. Annealing of amplicon from Loa isolate LRSW-3 was not included on this analysis due to its improper sequence length.

## 4. Discussion

Limited diversity and capabilities of soil protists to adapt to Atacama Desert extreme environments has been related to soil organic carbon availability in comparison with a higher number of mostly undescribed protists in aquatic communities [3,23]. Lithobiontic microbial consortia colonizing halites, quartz, and other rock habitats in Atacama rely on fog and salt deliquescence to overcome desiccation at one of the driest non-polar environments on Earth [9,10,11,23,24,25]. Then, microbial life, including protists, may proliferate successfully in habitats where liquid water is not a limiting factor.

Enclosed water bodies (Chiuchiu, Tebenquiche, and other ponds and lakes) and rivers (Loa, Salado, and others) can be found along the pre-Andean corridor and at the Andes plateau in the Region of Antofagasta [9,14]. Among them, the Loa River is a remarkable biogeographic icon of the Atacama Desert in northern Chile. It is the longest river in Chile, flowing along 440 km from its origin at the Andes Mountains, at 5650 m asl, across the Atacama hyperarid core to the Pacific Ocean. Near San Francisco de Chiuchiu village, the Loa River receives waters from its tributary, the Salado River, originating at El Tatio Geyser Field, over 4000 m asl. Substantial geological and hydrochemical analyses have been conducted in the Loa and Salado basins, but microbial studies are scarce, particularly on the presence of protists in their waters and sediments [26]. 

In nature, FLA from the genus *Acanthamoeba* affect the structure and abundance of microbial communities and are secondary decomposers influencing nutrient recycling with a positive impact on ecosystems and are resilient to harsh environmental conditions [1,5]. Thus, increasing efforts are needed to expand our knowledge of FLA presence, identification, distribution, diversity, and environmental niches along the northern and central hyperarid Atacama. 

What do we know about Atacama FLA? Information on the biogeographic distribution of FLA in the Atacama Desert is mostly limited to urban locations and at the southern border of Atacama. De Jonckheere et al. [15] mentioned *Naegleria* strain NG946 as an isolate from alpine Chile without further information on its original environmental setting. More recently, Fernandez et al. [7] reported on 416 taxa and 64 genera of testate amoebae in Chile. Astorga [5] identified *Acanthamoeba* in all 15 administrative regions in Chile, being predominant the T4 genotype, after analyzing over 500 samples that included vegetable, water, and soil, with general information on the sampling sites. In the Region of Antofagasta, only 2 samples from a total of 24 water and 5 soil samples were positive for *Acanthamoeba* [5]. In the shrub lands in the less hyperarid southern limit of the Atacama Desert, 29 taxa of testate amoebae species were described as the most abundant in soils under vegetation [27]; this observation agrees with a study on the Negev arid soils where shrubs provide appropriate niches for FLA feeding, reproduction, and diversity, particularly during the wet season, with Type 1 amoebas being the most abundant [28]. 

Our work on two major rivers in the pre-Andean corridor of the Antofagasta Region in northern Chile showed the dominant presence of members of the genus *Acanthamoeba* in water and sediment from the Loa and Salado rivers sampled at 2500 m asl. This is new information on novel FLA niches and on the biogeographic distribution of Atacama FLA. Future work will test our hypothesis that these Atacama members of the genus *Acanthamoeba* detected at Loa and Salado belong to different species and to the T4 genotype. Moreover, the identification of bacterial endosymbionts [2] opens new unexplored questions on Atacama FLA. Preliminary information (data not shown) indicates the presence of FLA members of the genera *Naegleria* and *Balamuthia* at other Atacama wetlands (Chiuchiu Pond near Salado River and Tebenquiche Pond at Salar de Atacama). These observations are an incentive to further characterize these new Atacama FLA and open new inquires on FLA living in the driest and oldest desert of our planet [29].

Finally, the importance of considering *Acanthamoeba* as a research model has recently been emphasized since these protists are reservoirs for human pathogens, sharing metabolic similarities with mammalian cells, and are Trojan horses for bacteria and viruses [30,31,32]. These reports provide recent insights on FLA as human health hazards and prompt new research activities on these protists. 

## Figures and Tables

**Figure 1 microorganisms-10-02315-f001:**
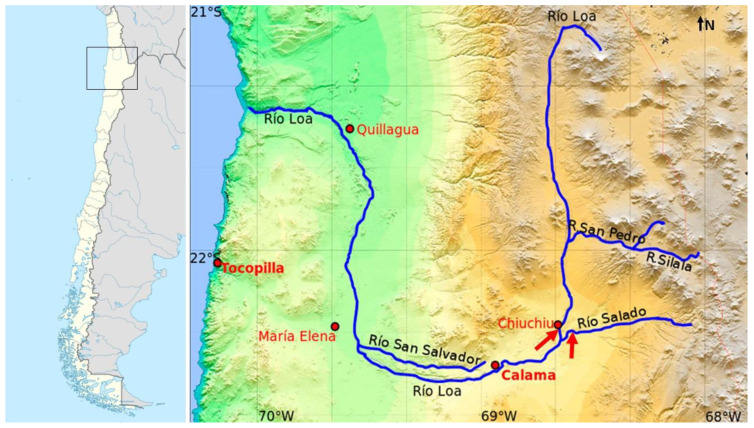
Partial view of the Atacama Desert at the Antofagasta Region, northern Chile. Sites at Loa and Salado rivers selected for sampling sites at pre-Andean corridor (nearly 2500 m above sea level) are shown by red arrows. Images were modified at https://en.wikipedia.org/wiki/Loa_River and https://es.m.wikipedia.org/wiki/Archivo:Chile_location_map.svg (accessed on 12 September 2022).

**Figure 2 microorganisms-10-02315-f002:**
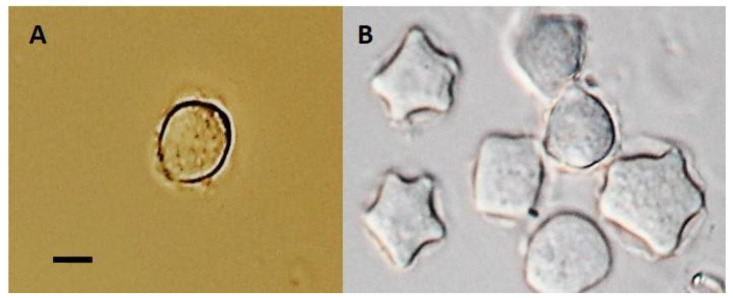
Light microscopy photographs of trophozoite (**A**) and cyst (**B)** stages of Atacama free-living *Acanthamoeba* sp. isolated from Loa River sediment sample. Magnification: 1000×. Bar: 10 µm.

**Figure 3 microorganisms-10-02315-f003:**
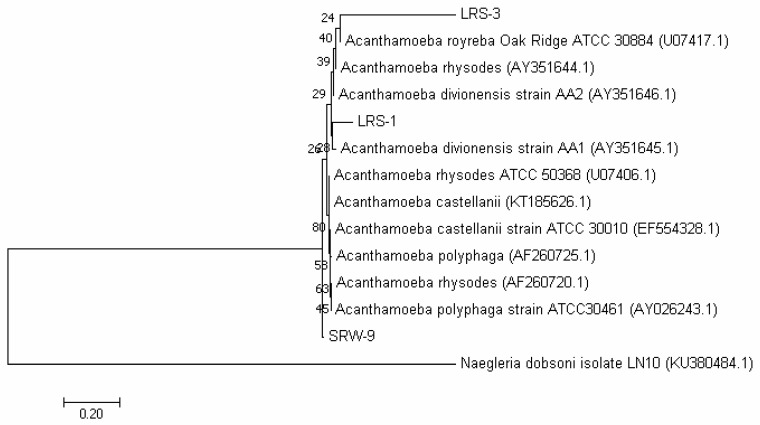
Phylogenetic analysis of Atacama *Acanthamoeba* spp. 18S rRNA gene sequences by the neighbor-joining method. Evolutionary analyses were done with MEGA7, and horizontal bars represent 0.50 substitutions per nucleotide site. The percentage of trees for associated taxa is shown at the branches with a bootstrap of 1000.

**Table 1 microorganisms-10-02315-t001:** Sampling sites, growth, and genus identification of free-living amoebae isolated from Salado and Loa rivers, Region of Antofagasta, northern Chile.

Sampling Site and Location	Sample ID	Description	Amoeba Growth	Genus
Loa River 22°20′18.9″ S 68°39′08.5″ W	LSR-1	Sediment	+	*Acanthamoeba*
	LRS-2	Sediment	−	
	LRS-3	Sediment	+	*Acanthamoeba*
	LRW-1	Water	+	*Acanthamoeba*
	LRW-2	Water	−	
	LRWS-3	Water with sediment	+	*Acanthamoeba*
Salado River 22°20′22.1″ S 68°35′56.3″ W	SRW-1	Water	+	*Acanthamoeba*

**Table 2 microorganisms-10-02315-t002:** Identification of Atacama FLA by BLASTn analysis. *Acanthamoeba* accession numbers from NCBI database are shown in parenthesis.

Sample ID	Primer	FLA	% Coverture	% Identity	Value-E
LRS-1 (Loa sediment)	PFLA	*Acanthamoeba* spp. MZOR (DQ103890.1)	99	89.70	0.0
		*Acanthamoeba rhysodes* (AY351644.1)	99	89.59	0.0
LRS-3 (Loa sediment)	JDP	*Acanthamoeba* genotype T4, islote IR-W34 (LC177665.1)	38	95.54	7e-41
		*Acanthamoeba* sp. T4, clone: RG-W6 (LC276365.1)	34	95.50	2e-40
LRS-3 (Loa sediment)	PFLA	*Acanthamoeba* spp. ACLCCOF-001 (EF205325.1)	82	100	3e-153
		*Acanthamoeba* spp. EFW4 (DQ992185.1)	82	100	3e-153
LRSW-3 (Loa water-sediment)	PFLA	*Acanthamoeba* spp. ALC2A (JQ271666.1)	92	100	1e-88
		*Acanthamoeba* spp. ALC10 (JQ271665.1)	92	100	1e-88
SRW-1 (Salado water)	PFLA	*Acanthamoeba* spp. OEW1 (AM412762.1)	100	99	0.0
		*Acanthamoeba* spp. PS (DQ185606.1)	99	95.74	0.0

## Data Availability

Sequences obtained in this work have been deposited at GenBank, NCBI, with the accessions OP297695 (isolate SRW-1), OP297696 (isolate LRSW-3), OP297697 (isolate LRS-3), and OP297698 (isolate LRS-1).

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
