# Peer review of "Presence of Free-living Acanthamoeba in Loa and Salado Rivers, Atacama Desert, Northern Chile"

_microorganisms, 2022, doi:10.3390/microorganisms10122315_

Round 1

Reviewer 1 Report

The authors have a deep knowledge of the Atacama desert microbiome, as witnessed by numerous publications on this topic.

The authors intend to increase knowledge about the presence, distribution, diversity, etc., of free-living amoebae in this extreme habitat. However, this objective seems too ambitious in view of the small number of samples collected. Few environmental samples collected in a single sampling session and in a very limited area (3 samples at a distance of 2 m from each other in the river Loa and 1 sample in the river Salado just 5 km away) are not representative of the ecosystem of the pre-Andean wetlands around Loa and Salado rivers. In my opinion, the small sample size is the main study limitation.

Minor revision

The section “sampling sites” must be rewritten because is difficult to understand the number and type of samples collected. The reference to the samples taken from Tebenquiche and Chiuchiu ponds confuses, because the results of these samples are not shown in the manuscript. The lines 64 and 65 are to be removed.

There are some abbreviations wrong (FLV or AFL instead of FLA).

In the table 2, the sample LRS-3 is present twice and LRW-1 is lacking.

Line 136. Write SRW-1 instead of SRW-9

The analysis of more water samples is needed in order to make the manuscript suitable for publication.

Author Response

I apologize to Reviewer 1 for my delay in answering the comments on my manuscript. I am just coming out of a very strong cold.

We understand and agree on Reviewer 1 comments and we also acknowledge his/her recognition on our previous work on the Atacama Desert microbiome. Our contribution has been considered as a Short Communication by the Editor handling our report sent to Microorganisms and we agreed on that, considering the data provided from our ongoing work.  Our goal is to demonstrate the presence of free-living amoebae in samples taken from two major rivers at the Antofagasta Region. We also recognize that our work includes a limited number of samples and sampling sites; however, our report will be the first contribution on the presence of free-living amoebae in these area of the Atacama Desert and future advances will be communicated according to our advances on this field.

We have considered all the minor revisions indicated (we apologize for these mistakes):

Section on sampling sites was edited (lines 58-67).

Wrong abbreviations have been modified.    

Sample LRS-3 (Table 2) is shown twice since it was challenged with two different primers

Sample LRW-1 was not included in Table 2 since it fails to provide results.

All modifications are included in the edited new manuscript.

Reviewer 2 Report

The article entitled "Presence of free-living Acanthamoeba on Loan an Salado rivers, Acatama desert, northern Chile" is a well writing manuscript that describes the presence of different free-living amoebae such as Acanthamoeba spp and Naegleria spp. at 2500 m above sea level. These amoebae are present in desert areas of Chile. In this way, the authors provide the knowledge that this type that these type of microorganisms are capable of inhabiting hyper arid places. 

Author Response

I apologize to Reviewer 2 for my delay in answering the comments provided. I am just coming out of a very strong cold.

We thank Reviewer 2 for the supporting understanding on our Short Communication. Our purpose was to share for the first time the existence of FLA in Atacama. We hope and we are sure that our ongoing research will provide new evidence on the presence of FLA along the Atacama Desert.

Reviewer 3 Report

This is a nice study and well written.

It will be nice if you can show the properties of 1 or 2 amoebae such as pathogenesis/ cytopathogenicity/ encystation assays and compare to a known strain such as the T4 genotype.

Also, it will be good to include Electron Microscopic images.

In the discussion, it will be good to discuss the role of Acanthamoeba as the trojan horse of the microbial world, and its role in the environment.

The following papers should be cited/discussed:

Rayamajhee, B., Willcox, M.D., Henriquez, F.L., Petsoglou, C., Subedi, D. and Carnt, N., 2022. Acanthamoeba, an environmental phagocyte enhancing survival and transmission of human pathogens. Trends in Parasitology.

Mungroo, M.R., Siddiqui, R. and Khan, N.A., 2021. War of the microbial world: Acanthamoeba spp. interactions with microorganisms. Folia Microbiologica66(5), pp.689-699.

Nageeb, M.M., Eldeek, H.E., Attia, R.A., Sakla, A.A., Alkhalil, S.S. and Farrag, H.M.M., 2022. Isolation and morphological and molecular characterization of waterborne free-living amoebae: Evidence of potentially pathogenic Acanthamoeba and Vahlkampfiidae in Assiut, Upper Egypt. PloS one17(7), p.e0267591.

Mungroo, M.R., Khan, N.A., Maciver, S. and Siddiqui, R., 2022. Opportunistic free-living amoebal pathogens. Pathogens and Global Health116(2), pp.70-84.

Author Response

I apologize to Reviewer 3 for my delay in answering the comments on our manuscript. I am just coming out from the very strong cold.

We agree that much more information and properties are needed on the FLA isolates from Atacama. The main purpose of our short communication  was to demonstrate the existance of FLA in Atacama. Our ongoing work is addressing such subjects and we hpoe we can provide more in-depth information on our next manuscript.

We have read and included the references suggested.

Round 2

Reviewer 1 Report

Dear authors, I appreciate your response and the corrections you have made. I agree with the editor’s proposal to publish your manuscript as a short communication. My only request is to add these words to the beginning of the sentence in line 186:

Although conducted on a limited number of samples and sampling sites, this study provides new information on...